# Circ-CPSF1 Worsens Radiation-Induced Oxidative Stress Injury in *Caenorhabditis elegans*

**DOI:** 10.3390/biom13010102

**Published:** 2023-01-04

**Authors:** Jing Yuan, Fei Lin, Zhiyong Wu, Zhilin Jiang, Ting Wang, Sitong Huo, Weinan Lai, Li Li, Chao Zhang

**Affiliations:** 1School of Traditional Chinese Medicine, Southern Medical University, Guangzhou 510515, China; 2Nanfang Hospital, Southern Medical University, Guangzhou 510515, China; 3Department of Biochemistry and Molecular Biology, Guangdong Provincial Key Laboratory of Single Cell Technology and Application, School of Basic Medical Science, Southern Medical University, Guangzhou 510515, China; 4The First School of Clinical Medicine, Southern Medical University, Guangzhou 510515, China; 5School of Medicine, Tsinghua University, Beijing 100084, China; 6Department of Rheumatology, Nanfang Hospital, Southern Medical University, Guangzhou 510515, China; 7College of Basic Medicine and Life Sciences, Hainan Medical University, Haikou 571199, China

**Keywords:** radiation, circ-CPSF1, DNA damage response (DDR), apoptosis

## Abstract

Radioactive substances have been used in various aspects in daily life. However, high-energy radiation could cause environmental problems, which would damage the human body. Circular RNA (CircRNA) has great potential in the minimization of ionizing radiation damage. To find a potential diagnostic and therapeutic target for reducing the damage of ionizing radiation, we selected circRNA cleavage and polyadenylation specificity factor subunit 1 (circ-CPSF1) based on its up-regulated expression after X-ray radiation and explored its effect on response to ionizing radiation using *Caenorhabditis elegans* (*C. elegans*). Circ-CPSF1 was screened out and its up-regulated expression was verified. The measurement of lifespan and germ cell apoptosis showed that circ-CPSF1 RNAi treatment extended lifespan and reduced apoptotic germ cells. ROS levels were significantly reduced after the interference of circ-CPSF1 in *C. elegans* with radiation. Mitochondrial membrane potential assay showed that the suppression of circ-CPSF1 could alleviate mitochondrial damage after radiation. Relative genes expression showed the involvement of circ-CPSF1 in radiation mediated DNA damage response pathways and apoptosis pathways. In conclusion, circ-CPSF1 exerts deleterious effects on lifespan, eggs production and germ cell apoptosis of *C. elegans* through oxidative stress, the DNA damage response (DDR) pathway, and the core apoptotic pathway after ionizing radiation, indicating the potential of circ-CPSF1 to be an important therapeutic target of radiation damage.

## 1. Introduction

Pollution caused by radioactive substances is called radiation pollution. The radioactive radiation environment is relevant to everyone [1]. Environmental and social problems caused by radiation have become more and more prominent [2]. The generation and origin of high-energy radiation pollution is very extensive. It is generally believed that the high-energy radiation pollution that endangers the living environment comes from five aspects, including waste from the atomic energy industry, fallout from nuclear weapons testing, medical inspections, scientific research, and construction and renovation materials [3]. The radioactive damage to the organism caused by radioactive pollution in the environment is mainly mediated by direct or indirect damage from rays to cellular biomolecule, such as DNA, lipids, and proteins [4,5]. Genome damage and instability has been observed in children after the Chernobyl nuclear accident [6,7]. Radiation-induced genome damage also was observed in families radiated by contaminated building materials in Taiwan and about 180 families were under radiation with a dose higher than 5 mSv/year [8]. In fact, radiotherapy has had a widespread application in the treatment of tumor and been an important source of high-energy radioactive pollution in the environment, which also brought the risk of toxicity and adverse effects to patients [9]. For example, the reproductive system is a sensitive target organ for ionizing radiation in the clinical treatment of tumor [10], which means that radioactive damage caused by radiotherapy could bring toxicity to the reproductive system, especially to the ovarian area. Ovarian toxicity is an important and common long-term side effect of radical radiotherapy, with long-term outcomes following radiotherapy showing reduced follicular reserve or ovarian atrophy [11]. Ionizing radiation during pregnancy may cause embryonic arrest, deformity, and even miscarriage [11]. Furthermore, it is well-documented that multiple organs and systems have an increasing risk of sickness and disease for radioactive damage produced in radiotherapy to patients and radiologic technologists, including the hematologic system, central nervous system, heart, lung, thyroid, breast, and skin [9,12]; evidence indicates that the breast cancer risk in female radiologic technologists significantly increases [13]. Thus, it is necessary to strengthen the protection against high-energy radioactive pollution in the environment and reduce its damage to the human body.

High doses of radiation cause severe damage to common model organisms, such as mice and zebrafish, which brings the challenge to the study of reproductive toxicity of high-energy radioactive pollution. Therefore, finding a suitable model organism is significant and would help overcome the challenge. In recent years, considerable research has indicated that *C. elegans* is an ideal tool to study radiation damage [14]. *C. elegans* has the ability to tolerate high doses of radiation, which would make it an excellent choice. Besides, common model organisms have long development cycles that complicate the study of their offspring. *C. elegans*, on the contrary, have a short life cycle and is a well-established biological model for research due to a series of characteristics such as easy observation, easy cultivation, highly conserved pathways [15], and numerous mutant strains [16] Furthermore, it has been widely utilized owing to its advantages of sensitiveness to environmental stresses [17].

CircRNAs are covalently closed RNA biomolecules that lack free 5′ and 3′ ends generated through back-splicing in eukaryotes [18]. CircRNAs are highly abundant and stable, usually expressed in tissue-specific or cell-specific patterns. Emerging evidence demonstrate that circRNAs could play critical roles in the occurrence and development of different types of diseases [19]. CircRNAs can act as micro RNAs sponge and material of translation, interact with proteins, form regulatory complexes with mRNA and directly regulate transcription [20]. It is reported that circRNAs affect some specific diseases in various ways referred to, such as pancreatic cancer [21]. CircITCH could also sponge miR-330-5p to upregulate the expression of SIRT6, Survivin, and SERCA2a, resulting in attenuating the effect of doxorubicin on injury and dysfunction of cardiomyocyte [22]. Macromolecules such as proteins, peptides, micro RNAs (miRNAs), and oligonucleotides are proven to be capable of regulating radiosensitivity. There are some evidences that circRNAs manipulate other macromolecules to regulate the radiosensitivity of diverse cancer, such as esophageal cancer, oral cancer, and lung cancer [23,24,25]. However, the long-term and overall effects of radiation damages has not been extensively studied, such as longevity and reproduction, which are closely related to body health. In *C. elegans*, massive circRNAs were significantly accumulated during aging, and expressions of some select circRNAs were surprisingly even up-regulated by over 40-fold [26]. Accumulating circRNAs may bring long-term effects to the body, which is possibly to be related to long-term radiation toxicity. Thus, circRNAs have potential value to be studied in researches of radiosensitivity and radioactive damages.

To further explore the role of circRNAs in ionizing radiation damage, we chose *C. elegans* as the animal model for our study. CPSF1, the host gene of circ-CPSF1, has high homology in humans, mice, nematodes, and other animals according to our result of phylogenetic tree. CPSF1 is the coding gene for Cleavage And Polyadenylation Specific Factor 1 protein. CPSF1 plays a crucial role in determining the specificity and efficiency of 3′-end processing of a subset of pre-mRNAs by recognizing the AAUAAA sequence [27]. CPSF1 performs different biological functions in various diseases. For example, the down-regulation of CPSF1 attenuates cell proliferation and enhances cell apoptosis [28]. However, CPSF1 has not been studied in the field of radiation, as well as its circRNA—circ-CPSF1. In this study, circ-CPSF1 was chosen to explore its role in radiation damage for its high expression after ionizing radiation and important function of the source gene. We specifically regulated the expression of circ-CPSF1 in *C. elegans* to study the effect and signal mechanism of circ-CPSF1 in ionizing radiation damage. The results of our research demonstrate that reactive oxygen species (ROS), DDR and apoptosis pathways are involved in circ-CPSF1 aggravating radiation damage. These indicate that circ-CPSF1 may be a new target for the exploration of early prevention and monitoring, which is crucial in high-energy radioactive pollution in the environment. Our research of the function and mechanism of circ-CPSF1 may provide a new idea for circ-CPSF1 as a diagnostic and therapeutic target of radiation injury in human.

## 2. Materials and Methods

### 2.1. C. elegans Culture

The N2 Bristol strain *C. elegans* (N2 *C. elegans*) (Caenorhabditis Genetics Center, Minneapolis, MN, USA) were grown at 20 °C in dark conditions in nematode growth media (NGM) which was made according to Luo’s method. The OP50 strain of *E. coli* (BioVector NTCC, Beijing, China) was fed to the worms according to a standard protocol [29]. HT115 (DE3) bacteria (MiaoLingBio, Wuhan, China) carrying the “empty” L4440 vector was fed to the worms as “L4440 group” which was control group. HT115 (DE3) bacteria carrying circ-CPSF1 recombinant plasmids and HT115 (DE3) bacteria carrying circ-DYN1 recombinant plasmids were also fed to the worms.

### 2.2. Synchronization of C. elegans

Synchronization was conducted according to standard methods [30]. First, adult worms when most of them were gravid were collected from plates with 1.5 mL of M9 buffer (3 g KH_2_PO_4_, 6 g Na_2_HPO_4_, 6 g NaCl, 1 mL 1 mol/L MgSO_4_ in 1 L deionized water and autoclave) (Aladdin, Shanghai, China) and were transferred to 15 mL tubes (NEST, Wuxi, China). Next, we add 3 mL sodium hypochlorite solution (0.5 mL 10 M NaOH, (Aladdin, Shanghai, China) 1 mL 20% bleach (Merck, Shanghai, China) into tubes. Then, the solution was vortexed for about 1 min, and was incubated for 3 min. The time point that most of worm bodies are dissolved was checked by a stereoscopic microscope (Motic, Richmond, BC, Canada). Then, eggs were spun down and washed 3 times with M9 buffer at once. The eggs were suspended with 3 mL M9 and were cultured overnight at 20 ℃. Transfer age-synchronized young worms to NGM plate to grow up.

### 2.3. Radiation Treatment

Our previous research have revealed a serious of novel ionizing radiation-response genes under high doses in *C. elegans* [31]. To discover new difference, we chose 50 Gy and 100 Gy as radiation doses. *C. elegans* in the larval 3–larval 4 (L3–L4) periods were exposed to 50 Gy and 100 Gy X-ray irradiation by a small animal irradiator (Faxitron, Tucson, AK, USA).

### 2.4. CircRNA Sequencing

Synchronized L4 *C. elegans* with and without exposure to 100 Gy ionizing radiation were cultured and used for high-throughput circRNA sequencing by Geneseed Biotech Co., Ltd. (Guangzhou, China). Total RNA was extracted and purified using a Magen Hipure Total RNA Mini Kit (Magen, Guangzhou, China). Then RNA sequencing libraries was constructed and a Qubit 3.0 fluorometer (Invitrogen, Carlsbad, CA, USA) was used for quality control. rRNA was removed and linear RNA was digested during library construction, specifically enriching circRNA. The HiSeq × 10 PE150 mode (Illumina Inc., San Diego, CA, USA) was used for circRNA sequencing. Differentially expressed circRNAs were screened.

### 2.5. Quantitative Real Time PCR

About 10,000 worms were collected into tubes and washed with M9 buffer 3 times. Trizol (Invitrogen, CA, USA) was used to isolate total RNA according to the manufacturer’s standard protocols. The qPCR reactions were performed using Hieff^®^ qPCR SYBR^®^ Green Master Mix (Low Rox Plus) (YEASEN, Shanghai, China) on real-time fluorescence quantitative PCR instrument (BiometraGmbH, Jena, Germany). The primer sequences are listed in Appendix A. The RNA relative expressions of the genes of interest were normalized using act-2 as a reference gene. The data was calculated using the comparative 2^−ΔΔCt^ method. Each comparison was performed using 9 Ct values from 3 biological replicates for relative expression of the target genes.

### 2.6. RNAi

Fragments specially targeting junction sites of circ-CPSF1 and circ-DYN1 were amplified from *C. elegans* genomic DNA. The following primers were used: circ-CPSF1 (forward, TCCACCGGTTCCATGGCTAGCCGAACATCGTCTCATTTG; reverse, GGGCCCCCCCTCGAGGTCGATCGGCTTCATCTGTAGGCAC); circ-DYN1 (forward, TCCACCGGTTCCATGGCTAGCACCAGGCATGAGACCACC; reverse, GGGCCCCCCCTCGAGGTCGAGACACCAGCACGCAAGAA). Next, L4440 vectors (MiaoLingBio, Hubei, China) were cut by NheⅠ (NEB, Ipswich, MA, USA) and SalⅠ (NEB, MA, USA) and DNA fragment was recovered. Add circ-CPSF1 segments or circ-DYN1 segments and L4440 segments up to 5 μL to produce circ-CPSF1 recombinant plasmids and circ-DYN1 recombinant plasmids. DNA segments mixture was incubated at 50 ℃ for 30 min with 5 μL aliquot of 2× Gibson master mix (NEB, MA, USA). Add 5 μL ligation product into 50 μL HT115(DE3) (MiaoLingBio, Wuhan, China). L4440 empty vector was transformed at the same time. Some single colonies could be seen on the LB solid medium (10 g tryptone, 5 g yeast extract, 10 g NaCl, adjust the pH to 7.0 with 1 M NaOH in 1 L of solution. Then 15 g of agar powder was added. (Aladdin, Shanghai, China) containing 50 µg/mL ampicillin (Macklin, Shanghai, China) overnight at 37 °C. Picked a single colony into 5 mL LB (10 g tryptone, 5 g yesat extract, 10 g NaCl, adjust the pH to 7.0 with 1 M NaOH and make up to 1 L) (Aladdin, Shanghai, China) and grew overnight at 37 °C until OD was 0.6. Seed NGM agar feeding plates (containing 50 µg/mL carbenicillin and 1mM IPTG (Macklin, Shanghai, China) with 200 µL of LB bacterial culture which included OP50, HT115 (DE3) bacteria carrying the “empty” L4440 vector, HT115 (DE3) bacteria carrying circ-CPSF1 recombinant plasmids and HT115 (DE3) bacteria carrying circ-DYN1 recombinant plasmids. Dry the plates overnight at room temperature. About 1000 worms were cultured on these four plates. Three days later, transferred about 1000 worms again to corresponding plates. The transfer was repeated 3 times.

### 2.7. Lifespan and Eggs Assay

The OP50 group, L4440 Group and circ-CPSF1 RNAi group worms in the larval 2 (L2) stage were treated with 50 Gy and 100 Gy X-ray irradiation. Then they were transferred to the 48-well plates. Thirty worms were included in each treatment. The worms were transferred to a NGM plate every 24 h until they stopped laying eggs at 20 °C. The number of eggs in each well was counted every day. No response to the stimulus of a tiny metal needle could be identified as the death of *C. elegans*. The time between L2 stage and death was noted as the lifespan of the *C. elegans*. Survival curves were plotted using Graphpad Prism 8 and compared by the log-rank (Mantel-Cox) analysis method. Lifespans were detected using 30 worms of each group. Total fecundity was the sum of the number of eggs laid per day.

### 2.8. Apoptosis Assay

Acridine Orange (AO) can be used to detect apoptotic germ cells in *C. elegans* [32]. In our study, L4 stage worms were exposed to 50 Gy and 100 Gy X-ray irradiation and were incubated with 100 mL of 25 mg/mL AO (TCI, Tokyo, Japan) at 20 °C for 1 h. The *C. elegans* were separated and washed by M9 buffer 3 times and moved to NGM plates for about 1 h. 10 worms were kept on the slides and anesthetized worms by 20 µL of sodium azide. The rate of cell apoptosis was evaluated by fluorescence microscopy (Nikon, Shanghai, China). Generally, uniform green fluorescence can be observed in the normal nonapoptotic gonad cells, while bright yellow or bright green fluorescence can be observed in apoptotic cells.

### 2.9. Detection of Reactive Oxygen Species Level

ROS levels were measured using H_2_DCF-DA (MCE, Shanghai, China) as a molecular probe. 200–300 L4 worms were kept in M9 buffer containing 5‰ DMSO (Beyotime, Shanghai, China) and OP50 in 1.5 mL tubes for each group. At the end of the exposure of 50 Gy and 100 Gy X-ray irradiation, *C. elegans* were moved into new tubes containing H_2_DCF-DA (Glpbio, Shanghai, China). The fluorescence was detected after 1 h under a fluorescence microscope (*n* = 10). ImageJ was used to measure relative fluorescent intensity.

### 2.10. Mitochondrial Membrane Potential Assay

Mitochondrial membrane potential (MMP) was measured by fluorescent probe JC-1(LEAGENE, Shanghai, China). *C. elegans* were soaked in JC-1 solution for 2 h in dark environment. M9 buffer was used to wash JC-1 dye on the surface of *C. elegans* in triplicate. Then, worms were anesthetized by 20 µL of sodium azide. The fluorescent intensity was observed by a fluorescence microscope and measured by ImageJ 1.8.0.

### 2.11. Statistical Analysis

Each experiment was performed in triplicate independently. All data shown in our research were means plus or minus standard deviations. IBM SPSS Statistics 20 or Graphpad Prism 8 was used to analyze statistics. Comparison of multiple groups was analyzed by one-way analysis of variance (ANOVA) followed by Dunnett’s *t*-test. A *p* value < 0.05 was considered significant and could be marked as * *p* < 0.05, ** *p* < 0.01, *** *p* < 0.001 and **** *p* < 0.0001.

## 3. Results

### 3.1. Characteristics of Differentially Expressed circRNAs Induced by Ionizing Radiation

To explore the differentially expressed circRNAs in ionizing radiation, both treated and untreated with 100 Gy X-ray irradiation Wild Type (WT) *C. elegans* were analyzed by high-throughput circRNA sequencing. We identified 36 upregulated and 37 downregulated circRNAs in ionizing radiation treated *C. elegans* compared to WT *C. elegans* controls (Figure 1a,b). Results of chromosome distribution analysis of differentially expressed circRNAs are shown in Figure 1c. The majority of differentially expressed circRNAs derived from chromosome 3, while circRNAs with no significant difference in expression were mainly derived from chromosomes 1 and 3. Accordingly, these results showed that the expression profiles of multiple circRNAs vary induced by ionizing radiation in *C. elegans*.

### 3.2. Circ-CPSF1 Was Upregulated in C. elegans Exposed to X-ray Irradiation and RNAi of Circ-CPSF1

To find potential circRNAs related to radiation damage, differential expression analysis was performed in synchronized *C. elegans* before and after exposure to 100 Gy ionizing radiation. As shown in Figure 2a, we selected 10 significantly differentially expressed genes from the sequencing results for further study. Junction site sequences of these 10 circRNAs were confirmed by PCR amplification with back-to-back specific primers. Chromas 2.6.6 was used to analyze the results of DNA-seq to verify these circRNAs (Appendix A). Next, RT-qPCR was used to further detect these circRNAs expressions in *C. elegans* exposed to ionizing radiation. Consequently, as presented in Figure 2b, circ-CPSF1, circ-DYN1, and circ-Y48A5A.1 were upregulated markedly after ionizing radiation. Circ-CPSF1 was especially most significantly changed among these 10 candidate circRNAs. The result of BLAST showed that circ-CPSF1 consists of the head-to-tail splicing of exon 10 and 11 of CPSF1 gene and the spliced junctions of circ-CPSF1 was verified by Sanger sequencing (Figure 2c). The phylogenetic tree of the CPSF1 protein family showed that CPSF1 has high homology in humans and *C. elegans* (Figure 2d). As circ-CPSF1 and circ-DYN1 were significantly upregulated after ionizing radiation, we hypothesized that circ-CPSF1 and circ-DYN1 were potential objects associated with ionizing radiation. To further investigate the effect of circ-CPSF1 and circ-DYN1 in *C. elegans* of ionizing radiation damage, we designed L4440 plasmids that specifically target the circ-CPSF1 and circ-DYN1 junction sites according to the circular structure of the circ-CPSF1 and circ-DYN1 molecules. The vector interference efficiency was detected by feeding bacteria containing L4440 vectors designed for *C. elegans* to evaluate the expression levels of circ-CPSF1 and circ-DYN1. Compared with the control group, the expression level of circ-CPSF1 and circ-DYN1 of RNAi groups were clearly reduced by more than 0.43 (*p* < 0.001) and 0.27 (*p* < 0.001) (Figure 2e). This data demonstrated that the RNAi plasmids could effectively interfere with the expression of circ-CPSF1 and circ-DYN1 in *C. elegans*.

### 3.3. Circ-CPSF1 Aggravated Radiation Damage to C. elegans on the Lifespan, Eggs Production and Germ Cell Apoptosis

As shown in Figure 3a, a highly significant lifespan shortening could be seen upon 50 Gy and 100 Gy ionizing radiation compared with non-irradiated controls. Moreover, the lifespan of the circ-CPSF1 RNAi group was prolonged after X-ray irradiation, while interference of circ-DYN1 greatly shortened the lifespan with or without irradiation. This indicates that circ-DYN1 is very likely to involve in other pathways rather than pathway of regulating radiation damage. Thus, we chose circ-CPSF1 for further study. In the natural growth state, no significant difference of the egg number existed between the OP50 group, the L4440 group and the circ-CPSF1 RNAi group (Figure 3b), which means that the two plasmids do not have any influence on egg production without radiation. The average egg production numbers of these three groups were 284.0, 279.6, and 284.9, respectively. After 50 Gy ionizing irradiation, the egg production of the three groups of nematodes was significantly reduced compared with the non-irradiated group, by 52.5%, 52.1%, and 48.8%. After 100 Gy ionizing irradiation, the number of eggs laid by the three groups of nematodes decreased sharply compared with the non-irradiated group, with a decrease of 98.2%, 98.2%, and 97.7%, respectively, and these nematodes were almost sterile. The number of eggs laid in the adult stage of nematodes decreased with the increase of radiation dose. However, after 50 Gy ionizing irradiation, the number of eggs laid in the RNAi group increased by 8.1% and 9.1% compared with the OP50 group and the L4440 group, respectively. After 100 Gy ionizing irradiation, the number of eggs laid in the RNAi group increased by 26.9% and 34.7% compared with the OP50 group and L4440 group, respectively. These results indicate that circ-CPSF1 could attenuate the protection of egg under ionizing radiation and this effect could be enhanced with higher dose of ionizing radiation. To determine how this effect occurs, we used AO to detect the apoptotic situation of germ cells. As shown in Figure 3c, the apoptotic germ cells appeared to be bright green. There was no significant difference in OP50 group (0.70 ± 0.64), L4440 group (0.30 ± 0.64) and circ-CPSF1 RNAi group (0.30 ± 0.64) without ionizing radiation. After exposure to 50 Gy ionizing irradiation, the number of germ cell corpses in circ-CPSF1 RNAi group (1.30 ± 0.64) greatly decreased compared to OP50 group (2.30 ± 0.64) and L4440 group (3.0 ± 1.09). Similarly, number of apoptotic cells in circ-CPSF1 RNAi group (2.70 ± 0.64) significantly decreased compared to OP50 group (4.30 ± 0.64) and L4440 group (4.60 ± 0.66). These mean that circ-CPSF1 could promote the apoptosis of germ cells with radiation and increase the number of apoptotic germ cells with the increase of dose. According to the results above, circ-CPSF1 aggravates radiation damage to *C. elegans* on the lifespan, eggs production and germ cell apoptosis.

### 3.4. The Induction of Circ-CPSF1 in Radiation Mediated ROS

It has been widely reported that ROS plays an indispensable role in apoptosis. Thus, we used H_2_DCF-DA as a superoxide probe to detect the ROS level in worms with and without X-ray exposure. With the increase of radiation dose, the fluorescence intensity of worms was markedly enhanced (Figure 4a), indicating ROS accumulates in *C. elegans* with radiation. Moreover, ROS level in worms treated with circ-CPSF1 RNAi was significantly reduced compared with that of the OP50 and L4440 groups with 50 Gy and 100 Gy irradiation. To further ensure the effect of circ-CPSF1 on ROS, we detected the expression of oxidative stress markers (sod-1, sod-3, ctl-2 and ctl-3). QPCR analysis showed that radiation significantly reduced the expressions of sod-1, sod-3, ctl-2 and ctl-3 while circ-CPSF1 interference could recover the levels of these antioxidant genes (Figure 4b). Thus, these results indicate that suppression of circ-CPSF1 could reduce ROS level and oxidative stress with radiation. However, the contrary result has been reported that radiation can induce some anti-oxidant genes in mice and *Caenorhabditis elegans* [33,34]. We noticed that ROS actually accumulate in these worms after radiation, which indicate that the radiation indeed injures the worms via oxidative stress. The regulation of antioxidant genes may depend on the dose and method of radioactive treatment. In some way, the upregulation of the genes means the self-adaption to radiation damages and the downregulation suggests the direct injury to gene. More evidence about the effects of radiation in different doses and ways are needed to figure out this contradiction. The effect of circ-CPSF1 interference to reduce oxidant stress in radiation is not relevant to the different outcomes. In fact, the decrease of the ROS and the recovery of expression of antioxidant genes in circ-CPSF1-RNAi compared with OP50/L4440 suggest that circ-CPSF1 participate in the process to impair antioxidant genes despite of the radioactive effects to the expression of these genes.

### 3.5. Function of Circ-CPSF1 in Mitochondrial Transmembrane Potential in C. elegans Exposed to X-ray Irradiation

It is well known that intracellular ROS production is closely related to mitochondrial damage. Mitochondrion represents a major source of cellular ROS generation. To ascertain the degree of contribution of mitochondria to ROS generation induced by radiation in three strains, JC-1, an ideal fluorescent probe, was used to measure mitochondrial membrane potential. JC-1 dye accumulates in mitochondria in a potential-dependent manner. In normal mitochondria, JC-1 aggregates in the mitochondrial matrix to form polymers, which emits strong red fluorescence; in unhealthy mitochondria, due to the decrease or loss of membrane potential, JC-1 can only exist in the cytoplasm in the form of monomers, resulting in green fluorescence. The change of JC-1 color can directly reflect the change of mitochondrial membrane potential. The degree of mitochondrial depolarization can be measured by the ratio of red/green fluorescence intensity. As shown in Figure 5, no significant difference in the ratios of red to green fluorescence was observed between the three strains without radiation, which means that there is no significant difference in mitochondrial membrane potential between the three groups. With radiation dose increasing, higher levels of JC-1 emitted more green fluorescence in three strains, indicating more severe mitochondrial damage. Of note was the ratio of red to green fluorescence of circ-CPSF1 RNAi *C. elegans* being evidently higher than OP50 and L4440 groups after 50 Gy and 100 Gy irradiation, respectively (Figure 5). The above results shows that interference of circ-CPSF1 is helpful for attenuating radiation damage to mitochondria, which indicates that ROS accumulation by circ-CPSF1 is very likely to be induced through radiation damage to mitochondria

### 3.6. Involvement of Circ-CPSF1 in Radiation Mediated DNA Damage Pathway and Apoptosis

To explore the underlying molecular mechanism of harmful effects of circ-CPSF1 after radiation, the mRNA expression of levels of DDR-related genes that positively upregulate DNA repair such as hus1, clk-2 and mrt-2 were measured by RT-qPCR. As shown in Figure 6a, the expressions of hus1, clk-2 and mrt-2 increased when treated with radiation in a dose-dependent way. Next, we explored the regulatory relationship between circ-CPSF1 and the apoptosis pathway. Apoptosis-related genes cep-1, egl-1, ced-4, ced-3, and ced-13 were upregulated after 50 Gy and 100 Gy X-ray irradiation. These genes in the OP50 group and L4440 group were significantly upregulated compared to that in circ-CPSF1 RNAi group. The apoptosis-inhibiting gene ced-9 of three groups was downregulated after 50 Gy X-ray irradiation (Figure 6b). Base on the results above, circ-CPSF1 possibly plays its role in radiation damage via DDR pathway and apoptosis pathway.

## 4. Discussion

At the United Nations Conference on Human Environmental Protection, radiation was listed as one of the main pollutions causing public nuisance [35]. The damage caused by radiation to the body increases with the increase of dose. Effective protection against high-energy radiation pollution in the environment is of great benefit to human health. The previous research results of our group shows that many genes are differentially expressed in *C. elegans* exposed to high-energy ionizing radiation [31]. It was reported that the difference in hatching rate insensitivity to X-ray irradiation is dependent on oogenesis [36]. Michael W. M. Jones et al. discussed dose-appropriate guidelines in the application of X-ray fluorescence microscopy for microscale-biological samples [37]. CircRNAs in radiotherapy have attracted increasing attention because of their potential biological functions [38]. Recent research shows that circRNAs can regulate tumor cell radiosensitivity via various pathways such as the Wnt/β-catenin pathway, PI3K/AKT signaling pathway and miRNA-sponging. However, no research has reported whether circRNAs involves in the impact of radiation on offspring. In the present study, we proved that interference of circ-CPSF1 could reduce radiation damage affecting the lifespan, eggs production and germ cell apoptosis of *C. elegans*.

In our study, we screened circ-CPSF1 as the target gene via circRNA sequencing and a series of biological experimental verification. CPSF1 has 4 homologous genes in *C. elegans*, Y76B12C.7a.1, Y76B12C.7b.1, Y76B12C.7c.1, and Y76B12C.7d.1. Circ-CPSF1 host gene Y76B12C.7a.1 was 32.22% identical with homo sapiens CPSF1 (Appendix A). Y76B12C.7a.1 was compared with sequences with other species with 93% bootstrapping (Figure 2d). The studies about circ-CPSF1 have not been reported. CPSF1 has been identified as a master cleavage and polyadenylation (C/P) factor in regulating alternative polyadenylation (APA) events. There is now emerging evidence that the depletion of CPSF1 attenuates cell proliferation, enhances apoptosis and causes cell cycle redistribution [39]. We found circ-CPSF1 is detrimental to the longevity and reproduction of *C. elegans* via affecting ROS production and DDR pathway in the context of radiation exposure, which might provide new insight for cancer radiation.

It has been confirmed that ionizing radiation accelerates aging [40]. In our study, we found lifespan shortening of *C. elegans* after radiation. Notably, the lifespan of RNAi group was longer than the OP50 group and L4440 group after radiation (Figure 3a). It was reported that in invertebrates reproduction is the most radiosensitive endpoint [41]. Radiation can affect progeny number, embryonic development, and germ cell apoptosis [42]. Indeed, our results showed significant decreases in the number of eggs after 50 Gy and 100 Gy of acute irradiation. Besides, the eggs of circ-CPSF1 RNAi group were more than the OP50 group and L4440 group, respectively (Figure 3b). Previous studies have shown that ionizing radiation can induce apoptosis in germ cells. Our findings not only are consistent with previous studies, but also show that circ-CPSF1 interference could reduce germ cell apoptosis (Figure 3c). For the first time, this data provides evidence that circ-CPSF1 aggravates the effects of radiation on lifespan, egg production and germ cell apoptosis in *C. elegans*.

Emerging studies showed that radiation exposure induces ROS production that initiates a series of molecular reactions, such as apoptosis [43]. To investigate the possible role of circ-CPSF1 on radiation, we detected the fluorescent signals of ROS in *C. elegans*. As shown in Figure 4a, ROS production in *C. elegans* exposed to 50 Gy and 100 Gy radiation was significantly increased, in a dose-dependent manner, compared with the OP50 group and L4440 group. We found that circ-CPSF1 interference significantly reverses levels of oxidative stress. The relative expressions of the antioxidant gene indicated that circ-CPSF1 could modulate the antioxidant enzymes to affect *C. elegans* exposed to ionizing radiation (Figure 4b). These results indicate that circ-CPSF1 promotes the accumulation of ROS in *C. elegans* when treated with X-ray.

Dysfunction of mitochondria is linked to the gradual production of mitochondrial ROS to a great extent [44]. To further evaluate the contribution of circ-CPSF1 to mitochondrial ROS generation which was induced by radiation, MMP change was performed using the fluorescent probe JC-1. Figure 5 shows that lower level of red fluorescence emitted by JC-1 with the increasing doses of radiation. Intriguingly, the proportion of red: green fluorescence was largely increased in RNAi group compared with the OP50 group and L4440 group after 50 Gy and 100 Gy radiation, respectively. These results suggest that circ-CPSF1 could aggravate mitochondria dysfunction in *C. elegans* when exposed to ionization damage. The overproduction of ROS mediates oxidate damages and impairs to mitochondria, which promote production ROS, forming a vicious cycle [45]. There have been evidences that circRNAs can change the functions of downstream biomolecules by sponging mi-RNAs and interacting proteins, then regulate the production of ROS [46,47]. Thus, we consider that circ-CPSF1 induces the overproduction of ROS with a vicious cycle in similar ways. It is noticeable that the mitochondria-located circRNAs have attracted more attentions for the tight association between ROS and mitochondrion. So, the location of circ-CPSF1 needs detection in subsequent study.

One of the important signaling cascade initiated by elevated ROS generation is the DDR pathway [48]. In *C. elegans*, mrt-2, hus-1, and clk-2 are well-known markers of DNA damage [49]. To examine the involvement of circ-CPSF1 in DDR pathway induced by radiation, firstly, we measured the impacts of X-ray irradiation on the expression of mrt-2, hus-1, and clk-2. All these genes expression levels increased after radiation. Furthermore, the untreated group and control group were compared and the expression levels of mrt-2, hus-1, and clk-2 were not so evident (Figure 6a). These data demonstrate that circ-CPSF1 was involved in the conserved DDR pathway induced by ROS. It has been commonly considered that downstream genes of DDR promote apoptosis. Besides, core apoptotic genes of *C. elegans* are some of the most important components of germ cell apoptosis, including cep-1, egl-1, ced-3, ced-4, ced-9, and ced-13. To further explain the major role of circ-CPSF1 in apoptosis induced by radiation, the expression of these genes was measured. As shown in Figure 6b, the OP50 group and L4440 group levels of cep-1, egl-1, ced-3, ced-4, and ced-13, but not ced-9, were significantly increased in comparison with the RNAi group after exposure to 50 Gy and 100 Gy X-ray, respectively. This research presents compelling evidence that circ-CPSF1 promotes the core apoptotic cascade pathway in germ cells of *C. elegans*. CircRNAs could interact with other macromolecules to perform their function, like RNA and protein. The circRNA-protein interaction patterns were even classified into changing interaction between proteins, binding or isolating proteins, introducing proteins into chromatin, formation of circRNA-protein-mRNA ternary complex and protein transfer or redistribution [50]. Various patterns of interaction provide circRNAs the ability to regulate other pathway in transcriptional and post-transcriptional ways. CircNR3C2 could sponge miR-513a-3p to promote HRD1-mediated tumor-suppressive effect in transcriptional way [51]. Unlike circNR3C2, CircMTCL1 was reported to promote complement C1q-binding protein (C1QBP)-dependent ubiquitin degradation to play its oncogenic function in laryngeal carcinoma [52]. Since circRNAs could perform their functions in different ways, whether circ-CPSF1 regulate the DDR and apoptosis related genes transcriptionally or post-transcriptionally, like mrt-2, is a valuable question to explore in further research.

For the duality of suppressing apoptosis to radiotherapy, the clinical benefits of circ-CPSF1 utilization are uncertain. The exploration about the effects of circ-CPSF1 to cancer patients suffering radiotherapy is needed. Other cancer model organism and cancer cell after radiation should been considered to complete the potential of circ-CPSF1. In other radiation damages, such as acute accidental radiation exposure, circ-CPSF1 can bring adequate and clear benefits. Besides, the long-term damages caused by radiation such as reproduction toxicity may be good indication.

In summary, our research indicates that circ-CPSF1 is closely associated with radiation in *C. elegans*. Circ-CPSF1 via ROS pathway, DDR pathway, and core apoptosis signaling pathway affects lifespan, egg production and germ cell apoptosis in *C. elegans* (Figure 7). Our study shows that circ-CPSF1 represents an important radiation damage therapeutic target whose interference can protect *C. elegans* from radiation. Considering the function of circ-CPSF1 in radiation damage, circ-CPSF1 has the great potential to be a diagnostic and therapeutic target in clinic and help reduce radiation pollution on human, especially radiation damage on ovarian.

## Figures and Tables

**Figure 1 biomolecules-13-00102-f001:**
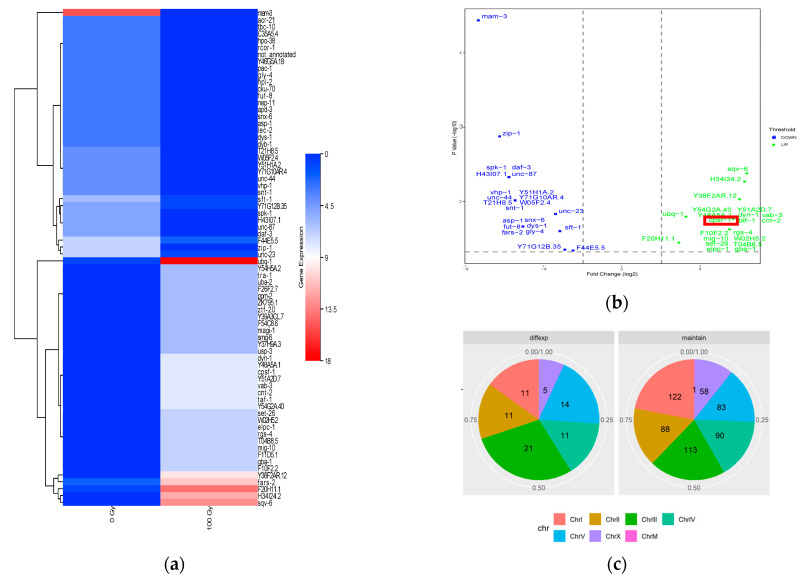
Characteristics of differentially expressed circRNAs induced by ionizing radiation. (**a**) Heatmap of all differentially expressed circRNAs in *C. elegans* after exposed to 100 Gy X-ray irradiation; (**b**) volcano scatter plot of differentially expressed circRNAs in *C. elegans* after exposed to 100 Gy X-ray irradiation. Circ-CPSF1 was marked by red square; (**c**) distribution map of circRNAs on chromosomes. The bands of different colors represent different chromosomal regions, and the numbers in different colors regions represent the numbers of circRNAs in different chromosomal regions.

**Figure 2 biomolecules-13-00102-f002:**
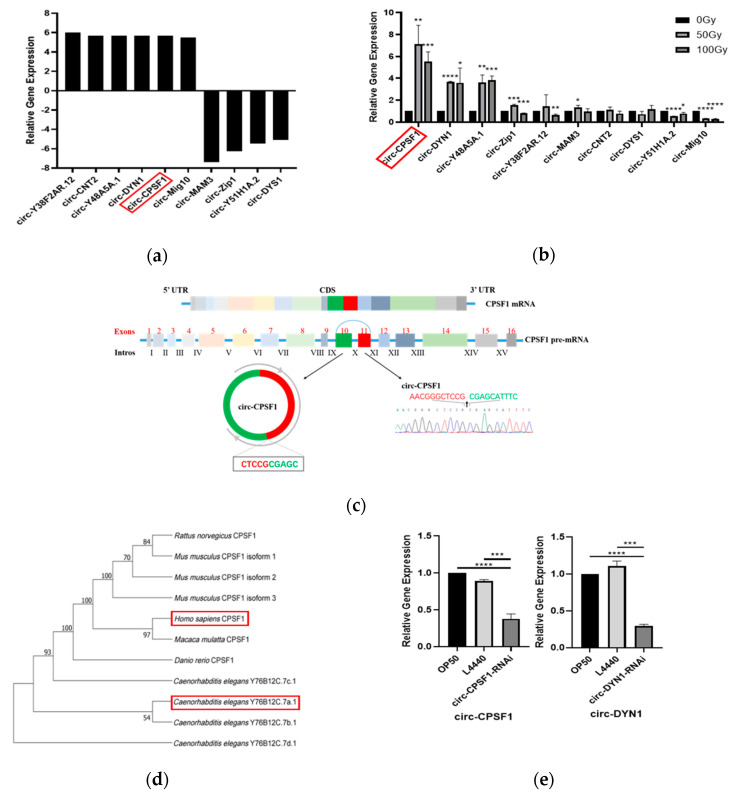
Circ-CPSF1 was upregulated in *C. elegans* exposed to X-ray irradiation and RNAi of circ-CPSF1. (**a**) The expression level of 10 circRNAs which include 6 upregulated and 4 downregulated circRNAs exposed to ionizing radiation; (**b**) RT-qPCR analysis of relative levels of 10 screened circRNAs exposed to 50 Gy and 100 Gy ionizing radiation; (**c**) illustration of genomic loci of circ-CPSF1, and the verification for the circ-CPSF1. Sanger sequencing was used to show the splicing site of circ-CPSF1; (**d**) the phylogenetic tree of the CPSF1 protein family. ClustalW was used for multiple sequence alignment. Bootstrap percentages are shown at each fork; (**e**) RT-qPCR analysis of the RNAi effect on circ-CPSF1 and circ-DYN1 expression in *C. elegans*. Circ-CPSF1 was marked by red square in (**a**,**b**,**d**). The results were presented as mean ± SD and were statistically analyzed by one-way ANOVA for gene expression. The statistical significance was represented by asterisks. * *p* < 0.05, ** *p* < 0.01, *** *p* < 0.001, **** *p* < 0.0001.

**Figure 3 biomolecules-13-00102-f003:**
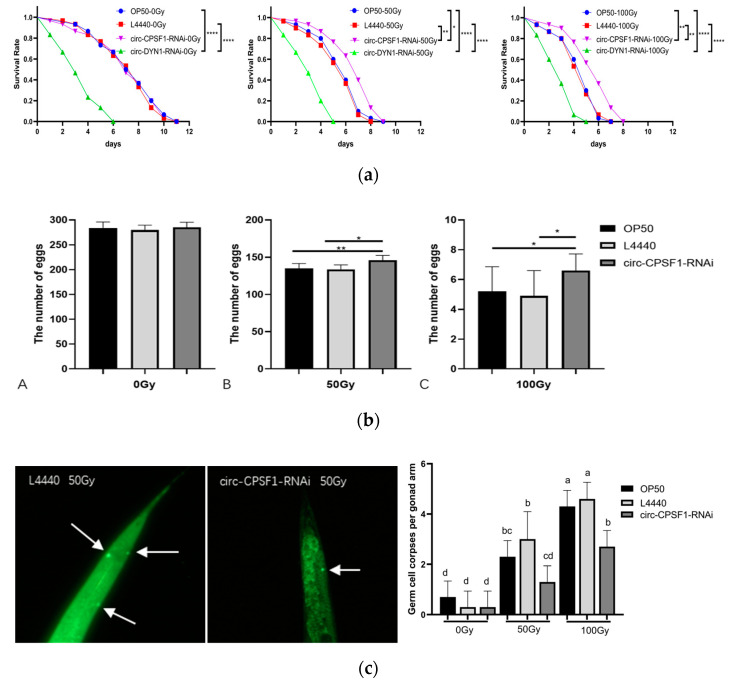
Circ-CPSF1 aggravated radiation damage to *C. elegans* on the lifespan, eggs production, germ cell apoptosis and embryogenesis. (**a**) Interference of the circ-CPSF1 prolonged the lifespan of the *C. elegans* after exposure to 50 Gy and 100 Gy ionizing radiation compared with OP50 group and L4440 group (*n* = 30). The results were presented as mean ± SD and the difference between different treatment groups was statistically analyzed by log-rank (Mantel-Cox) analysis. The statistical significance was represented by asterisks. * *p* < 0.05, ** *p* < 0.01, **** *p* < 0.0001; (**b**) interference of the circ-CPSF1 increased the eggs of the *C. elegans* after exposure to 50 Gy and 100 Gy ionizing radiation compared with OP50 group and L4440 group (*n* = 30); (**c**) after exposure to ionizing radiation, germ cells corpses per gonadal arm were detected (*n* = 10) under fluorescence microscopy (excitation: 485 ± 20 nm; emission, 520 ± 20 nm). Apoptotic germ cells are marked by white arrows. Different letters indicate significant differences at *p* < 0.05.

**Figure 4 biomolecules-13-00102-f004:**
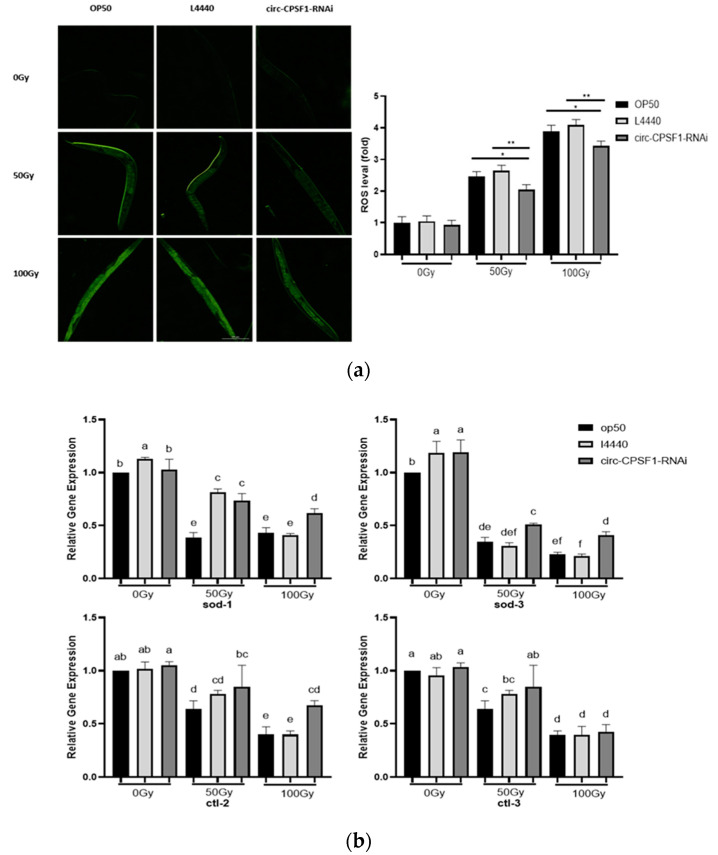
The induction of circ-CPSF1 in radiation mediated ROS. (**a**) The fluorescence images of ROS levels in three strains treated with radiation (*n* = 10). The relative fluorescence were quantified using the software Image-J; (**b**) RT-qPCR analysis of expression levels of oxidative stress related genes sod-1, sod-3, ctl-2, ctl-3 in three groups after 50 Gy and 100 Gy ionizing radiation. The results of ROS level and gene expression were presented as mean ± SD and were statistically analyzed by one-way ANOVA with Dunnett’s *t*-test (*p* < 0.05). Statistical significance was represented by asterisks and different lowercase letters. * *p* < 0.05, ** *p* < 0.01. Different letters in figure indicate significant differences at *p* < 0.05.

**Figure 5 biomolecules-13-00102-f005:**
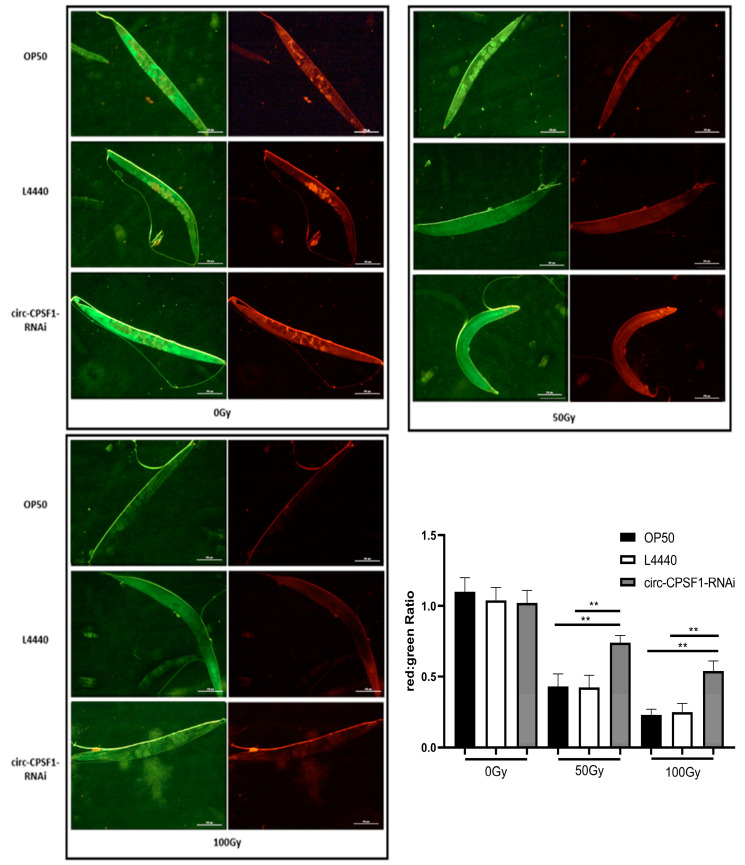
Function of circ-CPSF1 in mitochondrial transmembrane potential in *C. elegans* exposed to ionizing radiation. The florescent probe JC-1 was used to measure the mitochondrial transmembrane potential of *C. elegans* (*n* = 10). The relative fluorescence expression was measured using the software Image-J. The red: green fluorescence ratio of three groups were quantified. The results were presented as mean ± SD and were statistically analyzed by one-way ANOVA with Dunnett’s *t*-test (*p* < 0.05). Statistical significance was represented by asterisks. ** *p* < 0.01.

**Figure 6 biomolecules-13-00102-f006:**
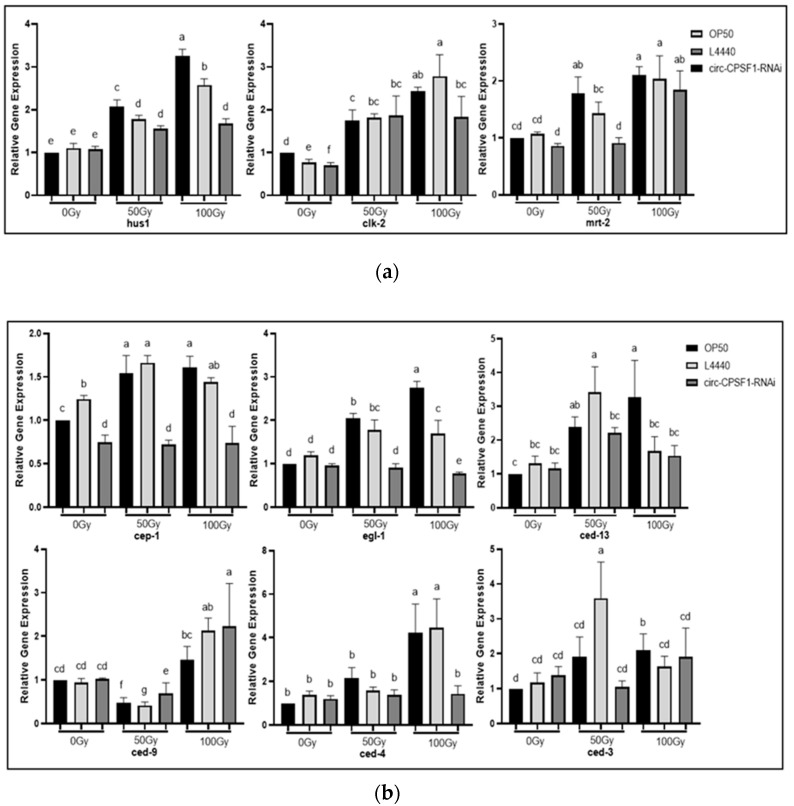
Involvement of circ-CPSF1 in radiation mediated DNA damage pathway and apoptosis. (**a**) RT-qPCR analysis of expression levels of DDR related genes hus1, clk-2 and mrt-2 in three groups after 50 Gy and 100 Gy ionizing radiation; (**b**) RT-qPCR analysis of expression of apoptosis related genes cep-1, egl-1, ced-13, ced-4, ced-9 ad ced-3 in three groups after 50 Gy and 100 Gy ionizing exposure. The results of gene expression were presented as mean ± SD and were statistically analyzed by one-way ANOVA with Dunnett’s *t*-test (*p* < 0.05). Statistical significance was represented by different lowercase letters. Different letters in figure indicate significant differences at *p* < 0.05.

**Figure 7 biomolecules-13-00102-f007:**
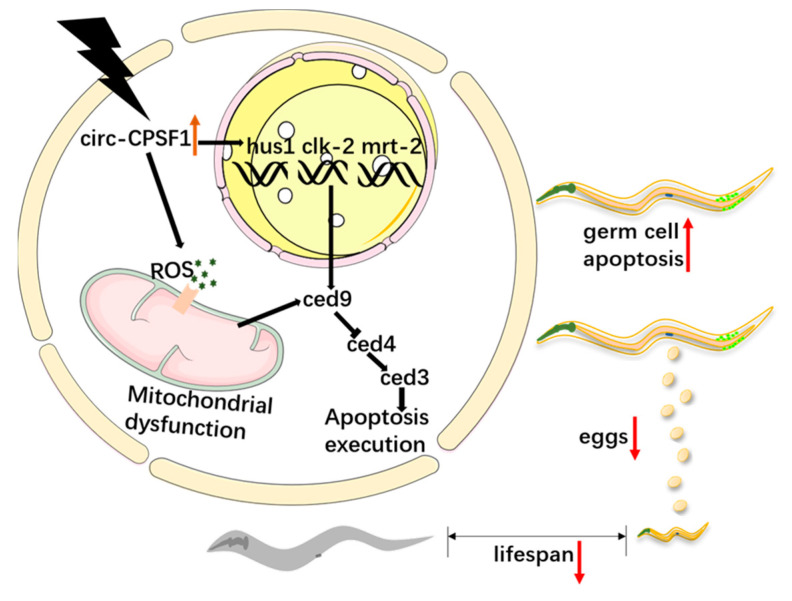
Model pattern of circ-CPSF1 regulatory network. Upregulated expression of circ-CPSF1 induced by ionizing radiation could impact lifespan, egg production and germ cell apoptosis of *C. elegans* and aggravated radiation damage via ROS, DDR, and core apoptosis pathways.

## Data Availability

Not applicable.

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
