# Peer review of "Circ-CPSF1 Worsens Radiation-Induced Oxidative Stress Injury in Caenorhabditis elegans"

_biomolecules, 2023, doi:10.3390/biom13010102_

Round 1

Reviewer 1 Report

This is a solid piece of research, well planned and conducted, although with  a couple of small questions on controls in my comments.

The heart of the paper is good but the introduction and discussion need significant revision. The introduction does not give a good overview of the cost of radioactive toxicity to the population. Side effects from radiotherapy are mixed interchangeably with radiation in the environment.

For the discussion, although there is a competent summary of results, this is a repeat of the results section. There is very little actual discussion of applied significance. I would suggest discussing; parallel examples of expression control by circ-RNA and mechanisms therefo, limitations to the study, future research direction and then how it might translate - presumably to a therapy for acute radiation exposure. The actual potential to redice later RT sequelae such as cancer would need future work also. although DNA repair of damaged cells is good, suppressed apoptosis might not be.

detailed comments are below:

Abstract

In general English syntax and sense correction is required.

Half of abstract is introduction and methods, suggest more results could go in too.

19 ‘radioactive radiation’. Is there another type?!. Just ‘radiation’ .

21 ‘in the field of ionizing radiation damage’ would be better stated as ’in the minimization of ionizing radiation damage.’

23  ‘used selected’ either word alone would do. ‘selected’ probably fits best.

25 ‘Circ-CPSF1 was screened by circRNA sequencing’ – does this make sense?

26 ‘Measurement of lifespan…’ rather than ‘detection’.

Introduction

In general I do not get a strong sense that the authors understand very well how radiation damages the body.

36 ‘radioactive radiation pollution’ – I don’t think this is an actual phrase, I cannot find it in searches and it doesn’t make sense, I can’t see how you can have radiation pollution that is non-radioactive. Just radioactive pollution OR radiation pollution.

44 The rays they emit will destroy the macromolecular structure in the body. This implies that there is widespread destruction of many macromolecules - Not sure this is true.

47 ‘radioactive substances caused by radiotherapy’ – radiotherapy causes radiation damage to DNA but does not generate ‘radioactive substances’ in the body.

51 Radiotherapy during pregnancy is avoided so this is not a justification for developing protection. Also it does not compute that, just because radiotherapy causes harm in pregnancy, that we therefore need protection against radioactive pollution in the environment. Authors should phrase justification better – statements quantitating the harm that environmental radiation can cause.

66 I would not think C elegans has high genetic homology with humans relative to other models described- fish and mice.

74 why does the accumulation of cicr RNAs in ageing give them potential value for radiosensitivity studies – this does not follow?

79 CPSF1 is the coding gene for Cleavage And Polyadenylation Specific Factor 1 protein. I note that circ-CPSF1 is explained in the results. Could the authors clarify this relationship earlier, in the introduction, to make the first part of the article easier to follow.

91 ROS needs defining here.

The last paragraph of introduction has some relatively detailed results in – suggest move elsewhere.

Methods

204 triplicate is three times.

Why did the authors pursue upregulated circ-RNAs but not those downregulated?

253 was CPSF1 protein level checked with the RNAi to make sure the effects displayed were not protein related (accepting that the technique aims to target the circ RNA).

Results

269 ‘As shown in Figure 3a, a highly significant lifespan shortening could be seen upon 50 Gy and 100Gy ionizing radiation’. In this figure the 0Gy, 50Gy and 100 Gy curves are on separate axes making comparison hard, suggest reorganize curves, prob a 1st curve just with radiation effect.

The fig 3b effect of circ-CPFS1 on egg numbers does not appear marked – especially the difference at 50gy where eggs appear to go from 140 to 130 or so but with a p < 0.0001. How was this calculation made – the effect does not look biologically significant. Apoptosis is more convincing in 3c.

RT has been shown to induce some anti-oxidant genes such as Sod1 elsewhere (eg https://doi.org/10.3109/09553002.2014.877174), authors should comment as they see a different outcome.

Fig 5 is somewhat dark although the bar graph effect is convincing. Can it be enhanced?

366 hus1, clk-2 and mrt-2 – would suggest mention that they positively upregulate DNA repair for clarity.

3.6 any idea of mechanism. Is it transcriptional control or post-transcriptional.

It would be good to do a quick experiment irradiating human cells and taking out the human equivalent to confirm this could translate to the human population.

Discussion

401 – any evidence of a circ-CPSF-1 of this type in humans? Suggest state if known or not.

408 ‘crucial for’ – should this read ‘detrimental to …. In the context of radiation exposure’.

411 – 459. The vast majority of this text nicely summarizes the results but they have already been relatively well-covered in the results section. There is little actual discussion or literature references in terms of the whether other circRNAs are involved, postulating mechanisms of control etc. Also discussing what experiments are needed going forward and in which scenarios this might be used for therapeutic benefit could be discussed eg after acute accidental radiation exposure. Finally, the cancer risk for either up- or down regulating circCPSF-1 would need to be explored- obviously enhancing DNA repair is good but suppressing apoptosis could be bad in this regard.

Author Response

Manuscript number: biomolecules-2014137

MS Type: Article

Title: circ-CPSF1 worsens radiation-induced oxidative stress injury in Caenorhabditis elegans

Jing Yuan , Fei Lin , Zhiyong Wu , Zhilin Jiang , Ting Wang ,

Sitong Huo , Weinan Lai * , Li Li * , Chao Zhang *Biomolecules

Correspondence Author: Weinan Lai * , Li Li * , Chao Zhang * 

Thank you very much for your consideration and reviewers’ comments concerning our manuscript entitled “circ-CPSF1 worsens radiation-induced oxidative stress injury in Caenorhabditis elegans” (Manuscript number: biomolecules-2014137). Those comments are valuable and very helpful for revising and improving our paper, as well as the important guiding significance to our research. We have studied the comments carefully and have made correction which we hope to meet with approval. Revised portions are marked in red in the manuscript. Simultaneously, the main corrections in the paper and the responds to the reviewers’ comments are also summarized and enclosed. If you have any questions, please do not hesitate to contact me.

Best wishes,

Chao Zhang, PhD

Department of Biochemistry and Molecular Biology, School of Basic Medical Science, Guangdong Provincial Key Laboratory of Single Cell Technology and Application, Southern Medical University

We sincerely thank the reviewer for providing constructive feedback and suggestions, which are of great value of help in improving our study. The following are our point-to-point responses to the specific comments of the reviewers.

Reviewer #1: This is a solid piece of research, well planned and conducted,

although with a couple of small questions on controls in my comments.

  1. The heart of the paper is good but the introduction and discussion need significant revision. The introduction does not give a good overview of the cost of radioactive toxicity to the population. Side effects from radiotherapy are mixed interchangeably with radiation in the environment.

Response: We sincerely thank the reviewer for the constructive and careful comment. According to the comment, we have revised the section of introduction about the radioactive toxicity. First, we corrected the mechanism of radioactive damage and described the side effects from radiotherapy as a significant radioactive source in environment to show radioactive toxicity to population. Then we described the radioactive ovarian toxicity to enhance the impression how radioactive toxicity impair human body. And we supplied other examples from Line 47 to Line 53 to quantitating the harm that environmental radiation can cause. We hope these descriptions can express the necessity to strengthen the protection against high-energy radioactive pollution in the environment and reduce its damage to the human body.Side effects from radiotherapy in introduction revised are the part of radiation in the environment.

  1. 2.In general English syntax and sense correction is required.

Response: We thank the reviewer for the constructive and careful comment. According to the comment, we have revised the manuscript and corrected mistakes.

  1. 3.Half of abstract is introduction and methods, suggest moreresults could go in too.

Response: We thank the reviewer for the constructive and careful comment. According to the comment, we have supplied more results from Line 27 to Line 30 in MS.

  1. 19 ‘radioactive radiation’. Is there another type? Just ‘radiation’.

Response: We thank the reviewer for the constructive and careful comment. We have revised the paper in whole MS.

  1. 21 ‘in the field of ionizing radiation damage’ would be better stated as ’in the minimization of ionizing radiation damage.’

Response: We thank the reviewer for the constructive and careful comment. We have modified..

  1. 23 ‘used selected’ either word alone would do. ‘selected’ probably fits best.

Response: We thank the reviewer for the constructive and careful comment. Modified.

  1. 25 ‘Circ-CPSF1 was screened by circRNA sequencing’ – does this make sense?

Response: We thank the reviewer comment. It has been modifiede.

  1. 26 ‘Measurement of lifespan…’ rather than ‘detection’.

Response: We thank the reviewer comment. According to related comments, we have modified in Line 26.

  1. 9.In general I do not get a strong sense that the authorsunderstand very well how radiation damages the body.

Response: We thank the reviewer for the constructive and careful comment. According to the comment, we have added some details from Line 47 to Line 56 and from Line 62 to Line 67 in MS. We will also strengthen the professional study on the radiation damages on the body.

  1. 36 ‘radioactive radiation pollution’ – I don’t think this is an actual phrase, I cannot find it in searches and it doesn’t make sense, Ican’t see how you can have radiation pollution that is non-radioactive. Just radioactive pollution OR radiation pollution.

Response: We thank the reviewer for the constructive and careful comment. It has beenmodified in Line 40 in MS.

  1. 44 The rays they emit will destroy the macromolecular structurein the body. This implies that there is widespread destruction ofmany macromolecules - Not sure this is true.

Response: We thank the reviewer comment. According to the comment, we have revised the paper from Line 47 to Line 49 in MS.

  1. 47 ‘radioactive substances caused by radiotherapy’ –radiotherapy causes radiation damage to DNA but does notgenerate ‘radioactive substances’ in the body.

Response: We thank the reviewer for the constructive and careful comment. According to the comment, we have revised the mechanism of radioactive and irrational expression from Line 62 to Line 67 in MS.

  1. 13.Radiotherapy during pregnancy is avoided so this is not ajustification for developing protection. Also, it does not compute that, just because radiotherapy causes harm in pregnancy, that we therefore need protection against radioactive pollution in the  Authors should phrase justification better – statements quantitating the harm that environmental radiation can cause.

Response: We thank the reviewer for the constructive and careful comment. According to the comment, we have revised the statement about the harm of the radiation form Line 47 to Line 56 and from Line 62 to Line 67. The explanation and more detailed responses are appeared in Q1 response.

  1. 14.66 I would not think C elegans has high genetic homology withhumans relative to other models described- fish and mice.

Response: We thank the reviewer for the constructive and careful comment. C. elegans model organism has been an excellent model to conduct microbiological experiments to research apoptosis, muscle atrophy, radiation effects, metabolic diseases, and aging, given its short life span compared to humans and other mammals. Although we have published several papers about the radiation damage in C. elegans model, the mechanism of circRNA in radiation damage is still unknown. According to the comment, we have revised our statement about C. elegans from Line 80 to Line 81. We believe the helpful suggestions make our study clearer and more reliable. Sincerely thank the reviewer again.

  1. 74 why does the accumulation of cicrRNAs in ageing give thempotential value for radiosensitivity studies – this does not follow?

Response: We thank the reviewer for the constructive and careful comment. According to the comment, we have revised our statements about circRNAs from Line 99 to Line 102. We believe the helpful suggestions make our study clearer and more reliable. Sincerely thank the reviewer again.

  1. 79 CPSF1 is the coding gene for Cleavage And PolyadenylationSpecific Factor 1 protein. I note that circ-CPSF1 is explained inthe results. Could the authors clarify this relationship earlier, in the introduction, to make the first part of the article easier to follow.

Response: We thank the reviewer for the constructive and careful comment. According to the comment, we have revised added a description about the relationship between CPSF1 and circ-CPSF1 from Line 105 to Line 106. We believe the helpful suggestions make our study clearer and more reliable. Sincerely thank the reviewer again.

  1. 91 ROS needs defining here.

Response: We thank the reviewer for the constructive and careful comment. According to the comment, we have added a definition of ROS in Line 116 in MS.

  1. The last paragraph of introduction has some relatively detailed results in – suggest move elsewhere.

Response: We thank the reviewer for the constructive and careful comment. According to the comment, we have revised the description of the results from Line 112 to Line 115 in MS.

  1. 204 triplicate is three times.

Response:We thank the reviewer for the constructive and careful comment. According to the comment, we have corrected the mistakes from Line 228 to Line 229 and in Line 232 in MS.

  1. 20.Why did the authors pursue upregulated circ-RNAs but not thosedownregulated?

Response: We thank the reviewer for the question. Differentially expressed circRNAs after ionizing radiation could be possible target to study the regulation of radiosensitivity. Thus, no matter the expression is upregulated or downregulated, we will explore their functions. We are also studying or going to study the downregulated circRNAs, and we believe that the results of these researches would be published in the future.

  1. 21. 253 was CPSF1 protein level checked with the RNAi to makesure the effects displayed were not protein related (acceptingthat the technique aims to target the circ RNA).

Response: We thank the reviewer for the question. The technique of RNAi aims to suppress the expression of circ-CPSF1 and circ-DYN1 but not the protein level of CPSF1 and DYN1. To explore the function of these two circRNAs for further research, we use qRT-PCR to verify the interference efficiency of RNAi plasmid after downregulating the expression of circ-CPSF1 and circ-DYN1. The idea whether the effect displayed was protein related is interesting and could be a potential further research for us to explore.

  1. 22. 269 As shown in Figure 3a, a highly significant lifespanshortening could be seen upon 50 Gy and 100Gy ionizing In this figure the 0Gy, 50Gy and 100 Gy curves are on separate axes making comparison hard, suggest reorganize curves, prob a 1 curve just with radiation effect.

Response: We thank the reviewer for the constructive and careful comment. After consideration, we decided not to change Figure 3a. The experiment of lifespan has 12 groups in total. If we plot all the curves of these groups, the figure would be difficult for readers to recognize the curves and compare the difference between effects of circ-CPSF1 and circ-DYN1 under the same dose. In order to focus the effect of circ-CPSF1 and circ-DYN1, we think that all the groups are needed in this experiment. Thus, we hope the result of lifespan could be showed as Figure 3a. If the reviewer still insists that the resulted should be presented in one plot, we would also accept this suggestion and redraw a new graph just as the reviewer described.

  1. 23.The fig 3b effect of circ-CPFS1 on egg numbers does notappear marked – especially the difference at 50gy where eggs appear to go from 140 to 130 or so but with a p < 0.0001. How was this calculation made – the effect does not look biologically  Apoptosis is more convincing in 3c.

Response: We thank the reviewer for the constructive and careful comment. After receiving the comments of the reviewer, we carefully checked our calculation and found out a mistake during calculating. According to the comment, we have corrected the result in the manuscript and updated Figure 3b.

  1. 24.RT has been shown to induce some anti-oxidant genes such asSod1 elsewhere (eg

https://doi.org/10.3109/09553002.2014.877174), authors should comment as they see a different outcome.

Response: We thank the reviewer for the constructive and careful comment. According to the comment, we have supplied our idea of the different outcome and added more discussion of this difference from Line 357 to Line 369 in MS.

  1. 25.Fig 5 is somewhat dark although the bar graph effect is Can it be enhanced?

Response: We thank the reviewer for the constructive and careful comment. According to the comment, we have enhanced the Figure 5 to make the outcome convincing and revise the statements. Please refer to the updated Figure 5. We sincerely thank the reviewer for considerate suggestions again.

  1. 366 hus1, clk-2 and mrt-2 – would suggest mention that theypositively upregulate DNA repair for clarity.

Response: We thank the reviewer for the constructive and careful comment. According to the comment, we have added a brief description of the effect of hus-1, clk-2 and mrt-2 from Line 409 to Line 410 in MS.

  1. 27.6 any idea of mechanism. Is it transcriptional control or post-transcriptional.

Response:We thank the reviewer for the constructive and careful comment. According to the comment, we have added our idea of the mechanism from Line 512 to Line 524 in MS. We sincerely thank the reviewer for considerate suggestions again.

  1. It would be good to do a quick experiment irradiating human cells and taking out the human equivalent to confirm this couldtranslate to the human population.

Response:We thank the reviewer for the constructive and careful comment. Because of some problems with radiation irradiation equipment, and we will arrange the related experiments in the future. 

  1. 29.401 – any evidence of a circ-CPSF-1 of this type in humans?Suggest state if known or not.

Response: We thank the reviewer for the constructive and careful comment. According to the comment, we have added a statement that circ-CPSF1 have not been studied before in Line 450 in MS.

  1. 408 ‘crucial for’ – should this read ‘detrimental to …. In thecontext of radiation exposure’.

Respone: We thank the reviewer for the constructive and careful comment. According to the comments, we have revised the context from Line 454 to Line 456 in MS.

  1. 411 – 459. The vast majority of this text nicely summarizes theresults but they have already been relatively well-covered in theresults section. There is little actual discussion or literature references in terms of the whether other circRNAs are involved, postulating mechanisms of control etc. Also discussing what experiments are needed going forward and in which scenarios this might be used for therapeutic benefit could be discussed eg after acute accidental radiation exposure. Finally, the cancer risk for either up- or down regulating circCPSF-1 would need to be explored- obviously enhancing DNA repair is good but suppressing apoptosis could be bad in this regard.

Response: We thank the reviewer for the constructive and careful comment. According to the comment, we have added more discussion from Line 488 to Line 495 and from Line 512 to Line 531 in MS.

Reviewer 2 Report

The present manuscript shows the role of circ-CPSF1 in the host damage radiation-response. Interesting, the authors demonstrated that this gene is associated to a severe damage response of the host when worms were exposed to radiation. Authors showed that the presence of circ-CPSF1 is associated to less lifespan, less eggs formation and high apoptosis in germ cells. However, it is necessary improved some aspects:

1. The evidence of the function of circ-RNAS is poorly described in the introduction. It is necessary increase the number of references in this point.

2. Are there more evidences about circ-RNAs and radiation damage response? I think that this point is poorly explained in the introduction.

3. The choice of radiation dose (50 or 100 Gy) must be explained in the text.

4. In the results section, figure 2d and 2c are referred before 2b. The order of the panel in figure 2 must be according the text.

5. The difference in lifespan between experimental groups OP50/L4440 and the circ-CPSF1-RNAi exposed with 50 and 100 Gy is discrete. Is relevant to the physiology of the worms a difference of 1 or 2 days?. 

6. The number of eggs observed in worms is 5-fold less in radiation-treated worms (100 Gy) compared with control (0 Gy). These differences are not relevant when the analysis is done with these two experimental conditions because differences are discrete in the high radiation condition. Therefore, differences between OP50, L4440 and circ-CPSF1-RNAi are not relevant because the three groups decrease the number of eggs in the same magnitude. So, according the data, is relevant the difference in the number of eggs between the groups in the high radiation exposure condition?.

7. The images of fluorescence microscopy are hard to analyze in figure 5. Please improve the quality of the images (increase brightness for example).

Author Response

Manuscript number: biomolecules-2014137

MS Type: Article

Title: circ-CPSF1 worsens radiation-induced oxidative stress injury in Caenorhabditis elegans

Jing Yuan , Fei Lin , Zhiyong Wu , Zhilin Jiang , Ting Wang ,

Sitong Huo , Weinan Lai * , Li Li * , Chao Zhang *Biomolecules

Correspondence Author: Weinan Lai * , Li Li * , Chao Zhang * 

Thank you very much for your consideration and reviewers’ comments concerning our manuscript entitled “circ-CPSF1 worsens radiation-induced oxidative stress injury in Caenorhabditis elegans” (Manuscript number: biomolecules-2014137). Those comments are valuable and very helpful for revising and improving our paper, as well as the important guiding significance to our research. We have studied the comments carefully and have made correction which we hope to meet with approval. Revised portions are marked in red in the manuscript. Simultaneously, the main corrections in the paper and the responds to the reviewers’ comments are also summarized and enclosed. If you have any questions, please do not hesitate to contact me.

Best wishes,

Chao Zhang, PhD

Department of Biochemistry and Molecular Biology, School of Basic Medical Science, Guangdong Provincial Key Laboratory of Single Cell Technology and Application, Southern Medical University

We sincerely thank the reviewer for providing constructive feedback and suggestions, which are of great value of help in improving our study. The following are our point-to-point responses to the specific comments of the reviewers.

Reviewer #2: The present manuscript shows the role of circ-CPSF1 in the host damage radiation-response. Interesting, the authors demonstrated that this gene is associated to a severe damage response of the host when worms were exposed to radiation. Authors showed that the presence of circ-CPSF1 is associated to less lifespan, less eggs formation and high apoptosis in germ cells. However, it is necessary improved some aspects:

  1. The evidence of the function of circ-RNAS is poorly describedin the introduction. It is necessary increase the number ofreferences in this point.

Response: We thank the reviewer for the constructive and careful comment. According to the comment, we have revised the statements about circ-RNAs and added more evidences and references about the function of circRNAs from Line 86 to Line 91 in MS.

  1. Are there more evidences about circ-RNAs and radiationdamage response? I think that this point is poorly explained inthe introduction.

Response: We thank the reviewer for the constructive and careful comment. According to the comment, we have supplied the evidence about circ-RNAs affecting radiation damage and radiosensitivity from Line 93 to Line 97 in MS.

  1. The choice of radiation dose (50 or 100 Gy) must beexplained in the text.

Response: We thank the reviewer for the constructive and careful comment to improve our manuscript. According to the comment, the reason why we chose 50Gy and 100Gy as radiation doses have been added from Line 145 to Line 147 in MS. Please refer to 2.3.

  1. In the results section, figure 2d and 2c are referred before 2b. The order of the panel in figure 2 must be according the text.

Response: We thank the reviewer for the constructive and careful comment. According to the comment, we have reordered the panel in Figure 2.

  1. The difference in lifespan between experimental groupsOP50/L4440 and the circ-CPSF1-RNAi exposed with 50 and 100Gy is discrete. Is relevant to the physiology of the worms adifference of 1 or 2 days?

Response: We sincerely thank the reviewer for the question. We have not done any study about the relevance between physiology difference of the worms with difference of 1 or 2 days. Considering that nematodes develop and differentiate quickly, we think that a difference of physiology might exist. Thank the reviewer for the question again. This question might be explored in the further research.

  1. 6.The number of eggs observed in worms is 5-fold less inradiation-treated worms (100 Gy) compared with control (0 Gy). These differences are not relevant when the analysis is done with these two experimental conditions because differences are discrete in the high radiation condition. Therefore, differences between OP50, L4440 and circ-CPSF1-RNAi are not relevant because the three groups decrease the number of eggs in the same magnitude. So, according the data, is relevant the difference in the number of eggs between the groups in the high radiation exposure condition?

Response: We sincerely thank the reviewer for the constructive and careful comment. As a living animal, C. elegans would move quickly and have other activities that cells do not have. Thus, inter-group error of the nematodes experiment certainly is not smaller than that of the cell experiment. Considering this difference between nematodes and cells, we have strictly controlled the counting time and repetition times when calculating results, so as to ensure the scientific results.

  1. The images of fluorescence microscopy are hard to analyzein figure 5. Please improve the quality of the images (increase brightness for example).

Response: We sincerely thank the reviewer for the constructive and careful comment. According to the comment, we have updated the images of fluorescence microscopy in Figure 5.
